

# Connecting regional aerosol emissions reductions to local and remote precipitation responses

Daniel M. Westervelt[1,2], Andrew J. Conley[3], Arlene M. Fiore[1,4], Jean-François Lamarque[3], Drew T. Shindell[5], Michael Previdi[1], Nora R. Mascioli[1,4], Greg Faluvegi[2,6], Gustavo Correa[1], Larry W. Horowitz[7]

[1]Lamont-Doherty Earth Observatory, Columbia University. Palisades, New York, USA
[2]NASA Goddard Institute for Space Studies, New York, New York, USA
[3]National Center for Atmospheric Research, Boulder. Colorado, USA
[4]Department of Earth and Environmental Sciences, Columbia University, Palisades, New York, USA
[5]Nicholas School of the Environment, Duke University. Durham, North Carolina, USA
[6]Center for Climate Systems Research, Columbia University, New York, NY, USA
[7]National Oceanic and Atmospheric Administration, Geophysical Fluid Dynamics Laboratory, Princeton, New Jersey, USA

*Correspondence to*: Daniel M. Westervelt (danielmw@ldeo.columbia.edu)

**Abstract.** The unintended climatic implications of aerosol and precursor emission reductions implemented to protect public health are poorly understood. We investigate the precipitation response to regional changes in aerosol emissions using three coupled chemistry-climate models: NOAA Geophysical Fluid Dynamics Laboratory Coupled Model 3 (GFDL-CM3), NCAR Community Earth System Model (CESM1), and NASA Goddard Institute for Space Studies ModelE2 (GISS-E2). Our approach contrasts a long present-day control simulation from each model (up to 400 years with perpetual year 2000 or 2005 emissions) with fourteen individual aerosol emissions perturbation simulations (160-240 years each). We perturb emissions of sulfur dioxide and/or carbonaceous aerosol within six world regions and assess the significance of precipitation responses relative to internal variability determined by the control simulation and across the models. Global and regional precipitation mostly increases when we reduce regional aerosol emissions in the models, with the strongest responses occurring for sulfur dioxide emissions reductions from Europe and the United States. Precipitation responses to aerosol emissions reductions are largest in the tropics and project onto the El Niño-Southern Oscillation (ENSO). Regressing precipitation onto an Indo-Pacific zonal sea level pressure gradient index (a proxy for ENSO) indicates that the ENSO component of the precipitation response to regional aerosol removal can be as large as 20% of the total simulated response. Precipitation increases in the Sahel in response to aerosol reductions in remote regions because an anomalous interhemispheric temperature gradient alters the position of the Intertropical Convergence Zone (ITCZ). This mechanism holds across multiple aerosol reduction simulations and models.

## 1 Introduction

Understanding the regional climate consequences of aerosols is of growing importance as emissions of aerosols and their precursors are projected to decline in most regions over the coming decades due to policies enacted to protect



human health from the negative effects of air pollution (Rao et al., 2017; van Vuuren et al., 2011). Global emissions of anthropogenic aerosols and their precursors, including sulfur dioxide ($SO_2$, precursor to sulfate aerosol), black carbon (BC), and organic carbon aerosol (OA) peaked in the 1970s and have been declining for the last few decades (Klimont et al., 2013; Smith et al., 2011; Smith & Bond, 2014). Major source regions such as the United States and Europe have also experienced

decreases in anthropogenic $SO_2$, BC, and OA emissions during this time (Leibensperger et al., 2012b; Tørseth et al., 2012). Emissions in China may also be beginning to decline, whereas emissions in India continue to increase (Fontes et al., 2017; C. Li et al., 2017; Lu et al., 2011). As emissions of anthropogenic aerosols are phased out regionally and globally, their removal is expected to affect global and regional precipitation (Shindell et al., 2012). However, we currently lack a full understanding of the magnitude, spatial pattern, statistical significance, and underlying physical mechanisms of the precipitation response.

In order to address this knowledge gap, we simulate here the precipitation responses to removal of aerosols from six world regions in three different fully coupled chemistry-climate models.

Aerosols impact precipitation primarily through two pathways: by altering the surface and top-of-atmosphere solar radiation balance (direct effect) and through microphysical effects on clouds (indirect effect) (Myhre et al., 2013). Generally, decreasing aerosol emissions results in a net enhancement of precipitation, since the reduced aerosol attenuation of incoming

solar radiation results in more radiation reaching the surface, thereby resulting in more available heat for evaporation and convection (Ramanathan et al., 2001; Rosenfeld et al., 2008). Additionally, aerosol removal may enhance autoconversion and thus further increase rainfall locally via the cloud lifetime effect (Albrecht, 1989), though this effect remains uncertain (Stevens & Feingold, 2009). Aerosol composition plays a role in determining precipitation response in both the direct and indirect pathways. Pure sulfate aerosols are scattering agents, while black carbon also absorbs incoming solar radiation, and

therefore may impact precipitation rates in different ways (Ming et al., 2010). Sulfate aerosols and some organic aerosols are efficient cloud condensation nuclei (CCN), while black carbon aerosols do not seed liquid clouds as readily (Bond et al., 2013; Petters & Kreidenweis, 2007), at least not initially. These differences in optical and microphysical properties related to aerosol composition may lead to unique precipitation responses to removal of individual aerosol components such as sulfate and black carbon (Andrews et al., 2010; Frieler et al., 2011).

Previous work has found that aerosols are linked to a number of regional precipitation and/or circulation responses, including location and width of the Intertropical Convergence Zone (ITCZ),  (Allen, 2015; Hwang et al., 2013; Ridley et al., 2015, Allen and Ajoku, 2016), rainfall in the Sahel (Ackerley et al., 2011; Biasutti & Giannini, 2006; Chang et al., 2011; Held et al., 2005; Rotstayn et al., 2002; Westervelt et al., 2017), South Asian monsoon circulation (Bollasina et al., 2011; Menon et al., 2002), phasing of the North Atlantic Oscillation (NAO) (Fischer-Bruns et al., 2009) and North Atlantic climate

variability (Booth et al., 2012), and rainfall in the US (Leibensperger et al., 2012b; Shindell et al., 2012). Additional work is needed to identify robustness across multiple models and understand physical mechanisms of these regional responses to aerosols. Westervelt et al. (2017) began this process by simulating the precipitation response to the complete removal of US



anthropogenic $SO_2$ emissions in three coupled chemistry-climate models and found statistically significant increases in Sahel rainfall in multiple models. We build here on the work of Westervelt et al. (2017) by considering, for the same three models, regional emissions removal from not only the US, but also Europe, China, India, South America, and Africa. Additionally, we expand the scope to include multiple aerosol types, including sulfate, BC, and OA in each of these regions. We identify

robust (and non-robust) precipitation responses to a variety of regional aerosol perturbations, and show that precipitation responses in the Sahel can be explained by a consistent physical mechanism involving a change in the interhemispheric temperature gradient and a northward shift of the ITCZ that is robust across multiple models. We choose to investigate the Sahel in more detail based on its recent climatic vulnerability to drought, which occurred over the latter half of the 20[th] century and was partially attributed to aerosol forcing (Ackerley et al., 2011; Biasutti & Giannini, 2006; Held et al., 2005).

Similarly, precipitation in the Mediterranean has declined since the mid-20[th] century, although the cause of this decline is not well understood (Giorgi, 2002; Xoplaki et al., 2004). While the response of Mediterranean precipitation to climate variability has been thoroughly investigated (Dünkeloh & Jacobeit, 2003; Krichak & Alpert, 2005), the potential role of aerosol forcing has not been examined. Therefore, we use our multimodel regional aerosol perturbation framework to focus on the Sahel and Mediterranean precipitation responses in detail in addition to our more general analysis of precipitation responses around the

globe.

## 2 Models and simulations

We use an identical modeling framework as described by Westervelt et al. (2017) and Conley et al. (2017). Briefly, we employ three coupled atmosphere-ocean-land-sea-ice climate models with fully interactive chemistry of aerosols and trace gases: 1) Geophysical Fluid Dynamics Laboratory Coupled Climate Model version 3 (GFDL-CM3) (Donner et al.,

2011), 2) Goddard Institute for Space Studies ModelE2 (GISS-E2-R) (Schmidt et al., 2014), and 3) Community Earth System Model version 1 (CESM1) (Neale et al., 2012). The model configuration for each is very similar to that used for the Coupled Model Intercomparison Project 5 (CMIP5). For further model description and model evaluation of relevance to precipitation response, we refer readers to Westervelt et al. (2017).

In each model, we conduct a series of long "present-day" control simulations of up to 400 years in length, forced

by perpetual year 2000 (2005 for NCAR-CESM1) conditions, including all emissions of aerosols and their precursors and greenhouse gas concentrations. We then conduct individual regional aerosol perturbation simulations in each model of at least 160 years and as long as 240 years, in which the anthropogenic aerosol or aerosol precursor emissions for a certain region are set equal to zero or reduced by the amount shown in Table 1. The magnitude of the emissions perturbation was chosen in order to have roughly equivalent emissions decreases across regions and models. As an example, "IN_$SO_2$" refers

to a simulation with perpetual year 2000 conditions (2005 for NCAR-CESM1), perturbed by setting all anthropogenic $SO_2$ emissions over India to zero. Other than the regional aerosol emissions perturbation, all other model settings remain identical to the control. Long control and perturbation simulations allow us to establish statistical significance and separate forced responses from internal climate variability. We also conduct an additional set of atmosphere-only, fixed-SST simulations of





40-80 years in length with control and perturbed aerosol emissions to calculate the effective radiative forcing (ERF, as defined in Myhre et al. (2013)) resulting from the regional perturbations to aerosol emissions.

### 3. Global precipitation responses to regional aerosol emissions reductions

5       Figure 1 presents the annual mean precipitation response to a given aerosol emissions perturbation in each of the three models for six different perturbation simulations. The remaining simulations are presented in Fig. S1 of the Supporting Information. Hatching represents statistical significance at the 95% level according to a simple Student's t-test. Each plot is the difference between the perturbation simulation and the control (e.g., US_SO$_2$ minus control), differenced at each exact month of the two simulations as done in Westervelt et al. (2017) and therefore can be interpreted as the precipitation

response to decreasing regional aerosol emissions. The first row (panels a through c) is for zero US SO$_2$ emissions and is discussed in detail in (Westervelt et al., 2017). Generally, across all perturbations, precipitation responses are largest in NCAR-CESM1, followed by GFDL-CM3 and GISS-E2. GISS-E2 simulations were performed in a setup that does not include a cloud lifetime effect (Schmidt et al., 2014), contributing to a smaller aerosol effective radiative forcing (Table 1 and Fig. S2), and a weaker precipitation response in that model. Global mean aerosol effective radiative forcing values for

each of the models are shown in Table 1, and a scatterplot of global mean precipitation changes versus global mean aerosol effective radiative forcing at the top of the atmosphere (TOA) is presented in Fig. S2. The aerosol ERF values are largest in NCAR-CESM1 followed by GFDL-CM3 and GISS-E2. Aerosol ERF is a factor of 2 or 3 smaller in GISS-E2 than in GFDL-CM3 and NCAR-CESM1, for some simulations. Overall, aerosol ERF is largest in NCAR-CESM1, ranging from 0 to 0.3 W m$^{-2}$ depending on the regional aerosol perturbation. Across the models, we find a strong-to-medium linear relationship between

global precipitation response and global effective radiative forcing in GFDL-CM3 (r = 0.70) and GISS-E2 (r = 0.5), but poor correlation in NCAR-CESM1 (r = 0.23). This suggests that TOA aerosol ERF may explain some of the variation in global precipitation response, but not all of it. Although global precipitation responses are known to be constrained by the atmospheric energy budget (Allen & Ingram, 2002; Ming et al., 2010, Liu et al., 2018), we find weaker correlation (e.g. r = -0.3 for GFDL-CM3) between global precipitation response and atmospheric absorption (TOA minus surface forcing) when

compared to global precipitation and TOA forcing alone. Samset et al. (2016) and Liu et al. (2018) found strong correlation between global precipitation "fast" response and atmospheric absorption. Their analysis correlated precipitation responses from fixed SST simulations with aerosol ERF, whereas our analysis in Fig. S2 correlates precipitation responses from coupled model simulations with aerosol ERF, which may explain the discrepancy. Recently, Chung and Soden (2017) showed that aerosol indirect effects could dominate precipitation responses to aerosol perturbations, consistent with our

finding that GISS-E2, lacking an aerosol cloud lifetime effect, has the smallest precipitation response.

      In Fig. 2, we present precipitation responses (perturbation minus control, representing aerosol decreases, as in Fig. 1) globally-averaged and averaged over two regions (which are shown in Sect. 4), the Sahel and the Mediterranean. Numbers





in the upper left of each panel of Fig. 2 represent the mean precipitation for the control run for each region and each time period. Figure 2 shows that the global precipitation responses (panel a) nearly always agree in the three models. In general, NCAR-CESM1 responds the strongest to aerosol decreases, with increases in global mean precipitation up to about 0.025 mm d$^{-1}$ or about 1% of the global mean in the control simulation. Global precipitation changes in GISS-E2 and GFDL-CM3

are similar in magnitude in many of the aerosol perturbation scenarios. Thirty-three of the thirty-four model simulations among the various regional emissions perturbations result in a global annual mean increase in precipitation, the one exception being US_BC in GFDL-CM3 (not statistically significant). In addition to heating the surface, BC removal results in cooling aloft in the free troposphere and an increase of shortwave radiation at the surface, both of which can drive convective updrafts and result in precipitation increases. This "fast response" of precipitation to BC reductions tends to

dominate the total response to BC, as shown in the Precipitation Driver Response Model Intercomparison Project (PDRMIP) results (Samset et al., 2016, Liu et al., 2018). Despite opposite-signed aerosol ERF (Table 1) between BC and sulfate perturbation simulations among the models, global precipitation responses are often in agreement in sign (e.g. EU_BC and EU_SO2 in NCAR-CESM1 and GFDL-CM3). Because of the surface heating influence of BC compared to the cooling effects of sulfate, previous research has shown that BC and sulfate perturb precipitation in opposite directions (Wang, 2007;

Ramanathan and Carmichael, 2008). Our results here, while still somewhat inconclusive, suggest that in some cases, BC emissions decreases may actually increase global and regional precipitation, similar to sulfate. This result highlights that the influence of BC on global precipitation is still largely uncertain (Pendergrass and Hartmann, 2012; Liu et al., 2018) with major knowledge gaps still remaining (Bond et al., 2013).

## 4. Connecting regional emissions to regional responses

### 4.1 Sahel (20 ºW – 40 ºE, 10 ºN – 20 ºN)

In the Sahel, we find mostly increases in mean wet season (June through September) precipitation due to removal of aerosol and precursor emissions for nearly all regional emission perturbation simulations and models. For example, in GFDL-CM3 and NCAR-CESM1, reducing US SO$_2$ emissions (Fig. 1 panels a and b), European SO$_2$ emissions (Fig. 1 panels d and e), Chinese SO$_2$ emissions (panels g and h), and US SO$_2$+BC+OC (panels m and n) induces a similar precipitation

increase over the Sahel. This indicates that decreasing aerosol and aerosol precursor emissions in places like the US, Europe, and China will increase rainfall over the Sahel by strengthening and shifting the northern edge of the ITCZ northward into the Sahel. This phenomenon is mostly not present in GISS-E2, which we partially attribute to the smaller aerosol forcing (ERF) and thus a smaller and insignificant (or nonexistent) interhemispheric temperature gradient (see Figs. 3 and 4 and associated discussion).

Figure 2b shows the change in wet season Sahel rainfall for all models and all simulations. Error bars indicate ±1 standard error of the mean. Out of the 34 model simulations conducted, only 9 show decreases in precipitation over the Sahel. Thus, we conclude that aerosol emissions decreases in regions around the world are likely to bring additional rainfall



to the Sahel. Similarly, our results agree with findings that aerosol and precursor emissions increases in the mid-20[th] century may have contributed to the mid-20[th] century Sahel drought (Biasutti & Giannini, 2006). The largest responses in Sahel rainfall occur in NCAR-CESM1, particularly in the US_SO$_2$, EU_SO$_2$, and US_ALL simulations, where increases in average wet season rainfall are as high as about 0.25 mm d$^{-1}$ or 10% compared to the control simulation seasonal mean. These

precipitation increases point to potential remote impacts of decreasing pollution in major emitting regions like the US and Europe, where emission reductions as a result of air pollution regulation may help reduce the likelihood and severity of future droughts in the Sahel. The models agree in the sign of the Sahel precipitation impact in 7 of the 12 perturbation simulations (only including the simulations that at least two models conducted). Small error bars in many of the simulations conducted with NCAR-CESM1 and GFDL-CM3 indicate statistical significance. We identify below a physical mechanism

that explains these increases, and show that it is consistent across multiple models and aerosol simulations.

Westervelt et al. (2017) and references therein argued that an anomalous warming in the Northern Hemisphere compared to the Southern Hemisphere due to removal of SO$_2$ emissions from the US produces a summertime (June-July-August) strengthening and a northward shift of the ITCZ, thereby delivering more wet season rainfall to the Sahel. We find a similar interhemispheric temperature gradient mechanism (defined as the difference between the entire northern

hemisphere and southern hemisphere temperature response to aerosol removal) in the EU_SO2 simulation (Fig. 3). Removal of European sulfur dioxide causes an anomalous heating of the Northern Hemisphere (+0.34 K versus 0.11 K in the Southern Hemisphere). The enhancement of the northern flank of the ITCZ and the accompanying northward shift is demonstrated in panel (b), which compares the control precipitation climatology (greyscale lines) to the responses (red-blue scale) over the Sahel. Furthermore, using the precipitation centroid method of Frierson & Hwang (2012), we find a northern shift of the

precipitation center of 0.1° latitude. Removal of either US or European aerosols results in strong anomalous warming of the Northern Hemisphere and thus precipitation enhancement in the Sahel.

In Fig. 4, we explore the robustness of this mechanism across our full set of regional aerosol emission perturbation simulations, and find that the change in Sahel wet season precipitation correlates with the change in interhemispheric temperature gradient induced by removing regional aerosol emissions in the GFDL-CM3 model (r = 0.89; Fig. 4, red

symbols). In other words, when the change in the interhemispheric temperature gradient is strongly positive in a given aerosol perturbation simulation—signifying anomalous warming of the Northern Hemisphere relative to the Southern Hemisphere--Sahel precipitation is enhanced. The notable exception to this is EU_BC, which causes a strong negative temperature gradient change due to the Northern Hemisphere cooling response from BC removal. When the gradient change is weak or even negative (e.g., EU_BC, upward triangle; IN_OC, diamond), precipitation in the Sahel slightly decreases due

to aerosol removal. The strength of the linear correlation illustrated in Fig. 4 suggests that the mechanism proposed in Westervelt et al. (2017) for US_SO$_2$ is robust for other regional aerosol emissions changes, and therefore is the dominant factor in GFDL-CM3 in explaining how regional aerosol emissions from remote regions around the world impact rainfall in the Sahel.



We find a similarly strong correlation in NCAR-CESM1 (r = 0.77; blue symbols, Fig. 4). This qualitative agreement between NCAR-CESM1 and GFDL-CM3 lends confidence to this mechanism of an anomalous Hadley cell circulation accompanied by a northward ITCZ shift that leads to Sahel rainfall increases when US and European aerosol emissions are reduced. In GISS-E2, there is no discernible interhemispheric temperature gradient in the response to Northern

Hemisphere aerosol emissions removal (Westervelt et al. 2017), and correspondingly, no statistically significant change in Sahel rainfall either (Fig. 4, green symbols). Although GISS-E2 differs from GFDL-CM3 and NCAR-CESM1, the non-response in precipitation and the lack of a change in the interhemispheric temperature gradient is consistent with our identified physical mechanism. The overall r-value combined across all three models is 0.70, indicating a robust relationship across the models.

**4.2 Mediterranean (20 ºW – 40 ºE, 10 ºN – 20 ºN)**

We show changes in wintertime (October through March) Mediterranean precipitation rates due to regional aerosol reductions in Fig. 2c. Aerosol decreases around the world mainly act to increase precipitation in the Mediterranean, with only 9 of the 34 model simulations resulting in precipitation decreases. The models agree on sign in 8 out of the 12 perturbation simulations in which at least two models were included. Locally, the European aerosol reduction simulations

(EU_$SO_2$, EU_all, EU_BC, and EU_OC) indicate enhanced precipitation in all models. In contrast, reductions of $SO_2$ emissions in the US lead to precipitation decreases over Europe in all models, with a substantial decrease indicated by NCAR-CESM1. However, reductions of other aerosol types in the US generally result in increases in Mediterranean precipitation. Our results point to a statistically significant role for aerosol forcing in contributing to drying and wetting trends in the Mediterranean. Error bars are generally larger in the Mediterranean than the Sahel (Fig. 2b), but are still small

enough to indicate statistical significance at the 95% confidence level for most of the simulations in NCAR-CESM1 and GFDL-CM3. The precipitation changes here are smaller in absolute and relative magnitude compared to the Sahel, with maximum precipitation increases for an individual perturbation simulation of about 0.04 mm d$^{-1}$ or 3.5% compared to the control simulation in GFDL-CM3. Shorter averaging periods over the peak rainy season (e.g. December and January) result in slighter larger precipitation increases of up to 5%.

We also seek to understand the statistically significant precipitation enhancement in Europe and the Mediterranean that appears in several of our simulations, particularly in GFDL-CM3. Figure 5 shows the wintertime changes (December through March) in sea level pressure (SLP) and near-surface winds (panel a) and precipitation (panel b) over Europe in GFDL-CM3. We find a strong, statistically significant north-south dipole pattern in SLP response to removal of European $SO_2$ emissions (EU_$SO_2$, Fig. 5), European black carbon aerosol emissions (EU_BC, Supplemental Fig S4), European

organic carbon emissions (EU_OC, Supplemental Fig S5), and all of the previous three types of European aerosol emissions combinecd (EU_ALL, Supplemental Fig S6). This results in a weakening of the prevailing westerlies and a southward shift of the storm track over the North Atlantic, leading to a drying in Northern Europe and a statistically significant wetting in



Southern Europe (Figs. 5 and S4-S6,). This pattern resembles a shift towards the negative phase of the North Atlantic Oscillation (NAO)—characterized by a weakened Iceland low and a weakened Azores high—which has been shown to bring drier conditions to Northern Europe and wetter conditions to Southern Europe and the Mediterranean (Hurrell, 1995; Visbeck et al., 2001). The mean climatological SLP pattern in GFDL-CM3 control simulation (not shown) is centered further

west than the anomalies shown in Fig. 5, suggesting that the effect of aerosols is also to shift the centers of action eastward. In NCAR-CESM1, we find that the removal of European aerosols results in an opposite north-south dipole response to GFDL-CM3 (supplemental Fig. S7) and little change in the centers of action. As a result, the precipitation response to European $SO_2$ removal (EU_SO$_2$) in the Mediterranean in NCAR-CESM1 is smaller than in GFDL-CM3, statistically insignificant, and not associated with weakened westerlies and a southward storm track shift. The precipitation response in

GISS-E2 to decreases in European $SO_2$ emissions is unique compared to the other two models, featuring neither a strong north-south dipole of SLP changes, nor a statistically significant Mediterranean precipitation response (Fig. S8). As evidenced by the different circulation responses in the North Atlantic among the models, the impact that aerosols may have on the North Atlantic circulation is not robust across models. However, North Atlantic SLP and precipitation responses within GFDL-CM3 are statistically significant and consistent across several different aerosol perturbation simulations. Our

results in GFDL-CM3 are consistent with findings in CAM3 (an older version of the atmospheric component of NCAR-CESM1) that show a positive NAO-like response to increasing aerosols (Allen and Sherwood, 2011). There is little contribution from ENSO to the precipitation response to aerosol removal in all simulations in all models (Sect. 5, Fig. 6) over the Mediterranean, suggesting that ENSO teleconnections cannot explain the modeled precipitation changes over this region.

**4.3 Other regions**

Reducing regional aerosol emissions also tends to cause statistically significant precipitation responses locally (i.e. in the emissions region). For example, all three models show increases in precipitation due to decreasing $SO_2$ over China (Fig. 1g, h, and i). These local impacts may be caused by microphysical factors, in particular enhanced autoconversion rates due to decreasing aerosols, causing further increases in rainfall locally. Local impacts are evident in the US_ALL, US_SO$_2$,

EU_SO2, and IN_SO2 simulations in all three models. These local precipitation responses tend to be weakest and statistically insignificant in GISS-E2, which is consistent with this model's omission of cloud lifetime effects. India BC decreases lead to either essentially no change or a small decrease in precipitation in India unlike $SO_2$, although these are not statistically significant and therefore cannot be distinguished from internal climate variability.

Figure S3 shows regional precipitation responses to all aerosol reductions scenarios in all models for three

additional regions: India (65 ºE - 90 ºE, 8 ºN-35 ºN), Eastern United States (95ºW – 70 ºW, 23 ºN – 50 ºN), and Eastern China (100 ºE – 130 ºE, 15 ºN – 50 ºN). In the Eastern US and Eastern China, the precipitation responses to changes in local aerosol emissions dwarf those from remote regions. The precipitation responses to regional aerosol emissions reductions in



the Eastern US and China are robust, however, with 28 of 34 and 23 of 34 simulations showing an increase in annual precipitation, respectively. Monsoon precipitation in India changes by up to 3-5% in GFDL-CM3 and NCAR-CESM1 in response to particular regional emissions reductions, but the sign of the change (increase or decrease) is inconsistent between models and simulations. Aerosol impacts on monsoon precipitation have been widely studied (Bollasina et al., 2011, 2014;

Lau & Kim, 2006; X. Li et al., 2015; Meehl et al., 2008; Menon et al., 2002; Song et al., 2014), and deeper analysis from our simulations is left for future work.

**5. The role of ENSO in the precipitation response to regional aerosol emissions reductions**

Figure 1 points to an ENSO-like (El Niño-Southern Oscillation) response in the tropical Pacific. In NCAR-CESM1 and GISS-E2, there is a strong east-west dipole response in the tropical Pacific, with drying to the west and wetting to the

east. These responses are some of the largest in any region, and are statistically significant in NCAR-CESM1. There are also significant impacts in the tropical Pacific in GFDL-CM3, especially in CH_SO$_2$ (panel g) and US_ALL (panel m), though the precipitation response is generally opposite in sign compared to NCAR-CESM1 and GISS-E2, with a wetting in the western tropical Pacific in GFDL-CM3 as opposed to a drying in the other two models.

We therefore extend our precipitation analysis by investigating the impact that aerosols may have on precipitation

through changes in the El Niño-Southern Oscillation (ENSO). To estimate the ENSO component of the precipitation response to regional aerosol emissions decreases, we first perform a linear regression of the monthly mean precipitation fields onto a monthly ENSO index at each grid point in the control simulation of each model. We use a large-scale Indo-Pacific zonal sea level pressure (SLP) gradient index representing Walker circulation variations, which are closely linked to ENSO (Vecchi et al., 2006). The Indo-Pacific SLP gradient is defined as the difference between regional average SLP in the

Indian Ocean/west Pacific (80ºE – 160ºE, 5ºS – 5ºN) and the central/east Pacific (160ºW – 80ºW, 5ºS – 5ºN). The index is computed for every simulation (control and perturbation) and differences in indices are calculated between each perturbation simulation and the control simulation. The ENSO component of the precipitation response to aerosol emissions reductions, $\Delta P_{ENSO}$ is then computed as:

$$\Delta P_{ENSO} = r_{P:ENSO} \Delta ENSO \qquad (1)$$

where $r_{P:ENSO}$ is the regression value (slope) between precipitation and the Indo-Pacific zonal SLP gradient index in the control simulation (one value per grid point) and $\Delta ENSO$ is the difference of the index between the perturbation and the control simulation.

The ENSO component of the precipitation response to aerosol emissions reductions is shown in Fig. 6, with the same layout as Fig.1 except with a smaller scale range (by a factor of 5). In each model, we find substantial responses mostly

in the tropical Pacific and Atlantic, with changes as high as 0.1 mm d$^{-1}$ or 20% of the total precipitation response (compare with Fig. 1). With the exception of the IN_SO2 simulation in all models and the IN_BC simulation in GISS-E2, the tropical



precipitation patterns in every simulation and their teleconnections in different parts of the world tend to resemble the positive phase of ENSO (El Niño). Though most of the large responses are in the tropics, there is some evidence of ENSO teleconnections, for example over the Amazon region in Brazil, where precipitation decreases (drying) typically occur in each simulation and each model associated with the positive phase of ENSO. The ENSO component of the precipitation

response is also apparent over the Indian monsoon region, manifested mostly as a drying, consistent with the positive phase of ENSO. Agreement between models is strongest for GFDL-CM3 and GISS-E2, which show similar $\Delta P_{ENSO}$ patterns for each of the different perturbation simulations in Fig. 6. All three models agree on a strong response in the US_$SO_2$ simulation; however, there is a weaker response in NCAR-CESM1 for the rest of the perturbation simulations compared to the other two models. Since the models each show ENSO-like responses in the tropical Pacific, albeit with varying degrees

of statistical significance and consistency, we conclude that no matter the emissions region or aerosol type, precipitation changes may occur via modulation of ENSO in the tropical Pacific as a result of aerosol decreases, and these changes mostly resemble the positive phase (El Niño).

### 6. Summary and conclusions

We conduct a series of fourteen aerosol emissions perturbation simulations (160-240 years each) in which we

perturb emissions of sulfur dioxide and/or carbonaceous aerosol within six world regions relative to a long present-day control simulation in three coupled chemistry-climate models: NOAA Geophysical Fluid Dynamics Laboratory Coupled Model 3 (GFDL-CM3), NCAR Community Earth System Model (CESM1), and NASA Goddard Institute for Space Studies ModelE2 (GISS-E2). We find local increases in precipitation near the source region for each individual aerosol perturbation (e.g., increases in Chinese precipitation for the CH_$SO_2$ simulation), with statistical significance mostly limited to two

models: NCAR-CESM1 and GFDL-CM3. We find strong tropical precipitation responses in all three models and in essentially all aerosol removal simulations. In NCAR-CESM1 and GFDL-CM3, a northward shift in the tropical North Atlantic ITCZ is associated with increased Sahel precipitation in several of the simulations in which aerosols are removed. Globally averaged, small increases in precipitation occur in nearly all (33 out of 34 simulations across the three models) aerosol emission removal simulations. Regional emissions removal of black carbon (BC) and sulfur dioxide alone both

increase global mean precipitation in some cases, despite opposite-signed ERF, highlighting the uncertainties remaining in BC aerosol impacts on precipitation. Precipitation response is weakest and largely lacks statistical significance in GISS-E2, partially attributed to the lack of a cloud lifetime effect and thus a weaker aerosol indirect effect, which was recently found to dominate tropical precipitation response to aerosols (Chung & Soden, 2017). Our results further support this conclusion, as we find the weakest radiative forcing and precipitation response in GISS-E2. Without sensitivity simulations that isolate

the cloud lifetime component of the precipitation response to regional aerosol emissions removal, howeer, it is difficult to determine conclusively whether cloud microphysical or large-scale dynamical mechanisms dominate the modeled precipitation response.



We estimate the aerosol effective radiative forcing (ERF) in each perturbation simulation in each model using a series of atmosphere-only simulations with sea surface temperatures fixed to present-day modeled climatological means. The global mean ERF values are positive in all model simulations with the exception of black carbon simulations (EU_BC, US_BC), and generally fall in the range of 0 to 0.3 W m$^{-2}$. ERF is largest in NCAR-CESM1, followed by GFDL-CM3 and

GISS-E2. In both GISS-E2 and GFDL-CM3, global precipitation response correlates strongly-to-moderately (r > 0.5) with global mean ERF, although NCAR-CESM1 shows a weaker correlation (r = 0.3). This implies that model differences in top of atmosphere (TOA) effective radiative forcing explain some, though not the majority of the variation in global precipitation response.

We further investigate the regional aerosol impact on remote precipitation and show a strong linear relationship

between the change in interhemispheric temperature gradient and changes in Sahel rainfall across all of the different aerosol emission perturbation simulations. Changes in the interhemispheric temperature gradient produce an anomalous Hadley cell circulation and an accompanying northward ITCZ shift, with implications for precipitation over the Sahel. This linear relationship holds across multiple models, suggesting that regional aerosol reductions impact precipitation via the same physical mechanism, which we interpret to be a large-scale dynamical response, across different models and different aerosol

perturbations. Higher latitude regional emissions reductions (e.g. US, Europe) lead to greater change in the interhemispheric temperature gradient and thus correspondingly larger changes in Sahel rainfall than lower latitude aerosol emissions perturbations (e.g. China, India, Africa, South America). Air pollution controls in Europe and the US may help reduce the likelihood and severity of future droughts in the Sahel, and by altering the interhemispheric temperature gradient can influence precipitation in regions far removed from the emission region.

We find increases in Mediterranean wintertime precipitation in two of three models in response to most aerosol removal perturbations, implying that increases in aerosols throughout the mid 20$^{th}$ century could have played a role in the observed decreasing precipitation trends. In GFDL-CM3, this precipitation response can be largely explained by an aerosol-induced weakening of the prevailing westerlies and southward shift in the storm track over the Atlantic. Despite the distinction between sulfate (and organic carbon) and black carbon as scattering (cooling) versus absorbing (warming)

species, respectively, we find that European precipitation, sea level pressure, and wind speed respond similarly to removal of each of these species emissions over Europe, implying a role for cloud microphysical effects in this local climate response. This mechanism, however, is not confirmed by either GISS-E2 or NCAR-CESM1, and is therefore not robust and requires future investigation. Previous work relating aerosol forcing to North Atlantic circulation (Chiacchio et al., 2011; Fischer-Bruns et al., 2009, Allen and Sherwood, 2012) has been limited to single models and results have been inconclusive across

studies. The impact of aerosols on the North Atlantic and Mediterranean climate (and the NAO) thus remains unclear, and may warrant additional work with a larger model ensemble, with the diagnostics needed to probe the driving mechanisms more deeply, and to assess robustness in a more rigorous manner.

El Niño-Southern Oscillation (ENSO) plays an important role in modulating the impact of regional aerosol removal on precipitation. We perform a linear regression analysis to determine the contribution of the ENSO component of the





precipitation responses to the total response to regional aerosol emissions. We find the ENSO component can be as large as 20% especially over the tropical Pacific, with teleconnections to South Asian monsoon precipitation and Amazon wet season rainfall. Regional aerosol emissions reductions tend to cause a shift to the positive ENSO phase (El Niño as opposed to La Niña), with a few exceptions. Model agreement on the ENSO component of the precipitation response is best for the

US_SO2 simulation, and best between GISS-E2 and GFDL-CM3.

Overall, our findings suggest that, despite large variations between different models, there are some robust precipitation responses to aerosol emissions that warrant future investigation with additional models to pursue even more robust estimates, ideally with uncertainty ranges, perhaps through model intercomparison projects such as the upcoming AerChemMIP (Aerosol Chemistry Model Intercomparison Project) (Collins et al., 2017). Other precipitation responses show

little consistency across the models, raising questions as to whether the model representation is insufficient to detect a role for aerosol emissions, or whether those responses are swamped by climate variability relative to any aerosol influence. Our analysis can serve as a benchmark for future efforts with fully coupled chemistry and interactive emissions within climate models that consider emissions perturbations from a broad sampling of regions and aerosol species.

**Code Availability**

The code for the atmospheric component of the GFDL-CM3 model is available here: https://www.gfdl.noaa.gov/am3/. NCAR-CESM1 model code is available here: http://www.cesm.ucar.edu/models/cesm1.0/. GISS-E2 model code is available here: https://simplex.giss.nasa.gov/snapshots/.

**Data Availability**

Model data has been made available through the Figshare repository. NCAR-CESM1 data is available here:
https://figshare.com/articles/CESM1_precip/5738568 and here https://figshare.com/articles/ERF/5732397. GISS-E2 data is available here: https://figshare.com/articles/GISS_data/5738565/1, and GFDL-CM3 is available here: https://figshare.com/articles/GFDL_precip_data/5738562. Model data is also available on the high performance computing clusters for each of the modeling centers. Contact the corresponding author for any additional data requests.

**Acknowledgements**

Funding for this study was provided by an NSF EaSM-3 grant AGS 14-19398. The authors declare no conflicts of interest, and views, opinions, and findings presented in this paper are solely those of the authors and do not reflect the views of the funding agency. The NCAR-CESM work is supported by the National Science Foundation and the Office of Science (BER) of the U.S. Department of Energy. NCAR is sponsored by the National Science Foundation. GISS-E2-R simulations used resources provided by the NASA High-End Computing (HEC) Program through the NASA Center for Climate
Simulation (NCCS) at Goddard Space Flight Center.



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



**Table 1: List of aerosol perturbation simulations, emissions reductions relative to the control simulation, and corresponding aerosol effective radiative forcing (ERF). $SO_2$ = sulfur dioxide, BC = black carbon, OC = organic carbon aerosol, ALL = $SO_2$+BC+OC, BB = biomass burning, US = United States, EU = Europe, CH = China, IN = India, AFR = Africa, SA = South America. N/A means that the particular simulation was not performed with this model. "Zero" refers to a zero-out of emissions, 80% refers to an 80% reduction.**

| | GFDL-CM3 | | | NCAR-CESM1 | | | GISS-E2 | | |
|---|---|---|---|---|---|---|---|---|---|
| Simulation name | Type | Emis. (Tg species yr$^{-1}$) | ERF (W m$^{-2}$) | Type | Emis. (Tg species yr$^{-1}$) | ERF (W m$^{-2}$) | Type | Emis. (Tg species yr$^{-1}$) | ERF (W m$^{-2}$) |
| US_$SO_2$ | Zero | 14.8 | 0.16 | Zero | 14.0 | 0.14 | Zero | 14.8 | 0.056 |
| US_BC | Zero | 0.37 | -0.013 | Zero | 0.4 | 0.11 | | N/A | |
| US_OC | Zero | 0.82 | -0.008 | Zero | 0.8 | 0.12 | | N/A | |
| US_ALL | Zero | 14.8 $SO_2$ 0.37 BC 0.82 OC | 0.14 | Zero | 14.0 $SO_2$ 0.4 BC 0.8 OC | 0.23 | Zero | 14.8 $SO_2$ 0.36 BC 0.68 OC | 0.068 |
| EU_$SO_2$ | 80% | 14.6 | 0.18 | Zero | 18.3 | 0.18 | Zero | 18.6 | 0.09 |
| EU_BC | Zero | 0.77 | -0.095 | Zero | 0.8 | -0.03 | | N/A | |
| EU_OC | Zero | 2.63 | 0.026 | Zero | 2 | 0.15 | | N/A | |
| EU_ALL | 80% zero zero | 14.6 SO2 0.77 BC 2.63 OC | 0.13 | | N/A | | | N/A | |
| CH_$SO_2$ | 80% | 14.2 | 0.089 | Zero | 15.1 | 0.12 | 80% | 14.3 | 0.041 |
| IN_$SO_2$ | Zero | 5.7 | 0.13 | Zero | 5.6 | 0.11 | Zero | 5.63 | 0.037 |
| IN_BC | Zero | 0.54 | -0.038 | Zero | 0.6 | 0.06 | Zero | 0.53 | 0.011 |
| IN_OC | Zero | 2.78 | -0.024 | | N/A | | | N/A | |
| AFR_BB | 33% | 0.41 $SO_2$ 0.41 BC 5.3 OC | 0.026 | Zero | 0.4 SO2 0.4 BC 3.3 OC | 0.10 | Zero | 1.24 SO2 1.22 BC 12.5 OC | 0.108 |
| SA_BB | Zero | 0.40 $SO_2$, 0.40 BC 4.7 OC | 0.026 | Zero | 0.40 SO2 0.40 BC 3.3 OC | 0.34 | Zero | 0.41 SO2 0.41 BC 4.6 OC | 0.077 |




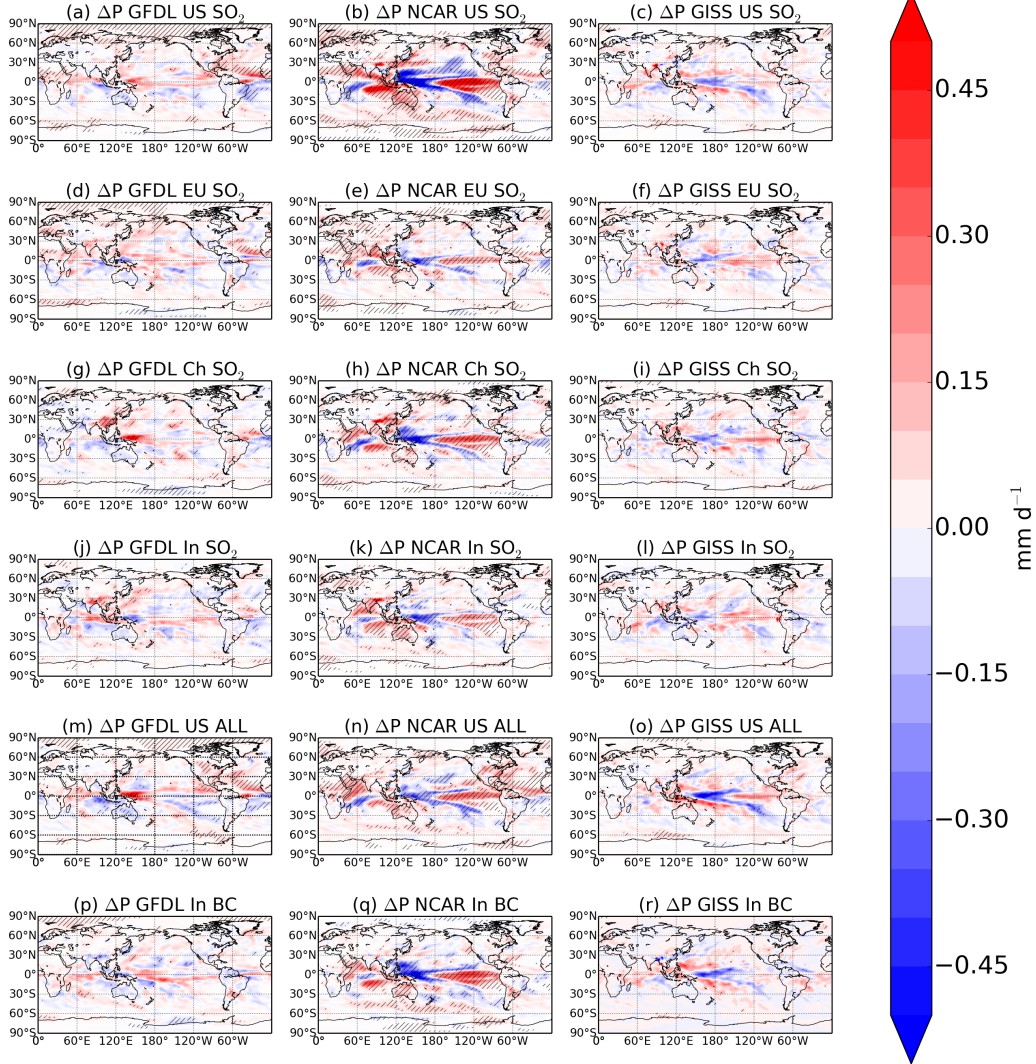

**Figure 1: 200-year annual mean precipitation response to aerosol emissions decreases in each of the three models (GFDL-CM3, first column; NCAR-CESM1, second column; GISS-E2, third column) for several different regional emissions decreases (simulations indicated in figure titles; see Table 1). Hatching represents statistical significance at the 95% level according to a Student's t-test.**





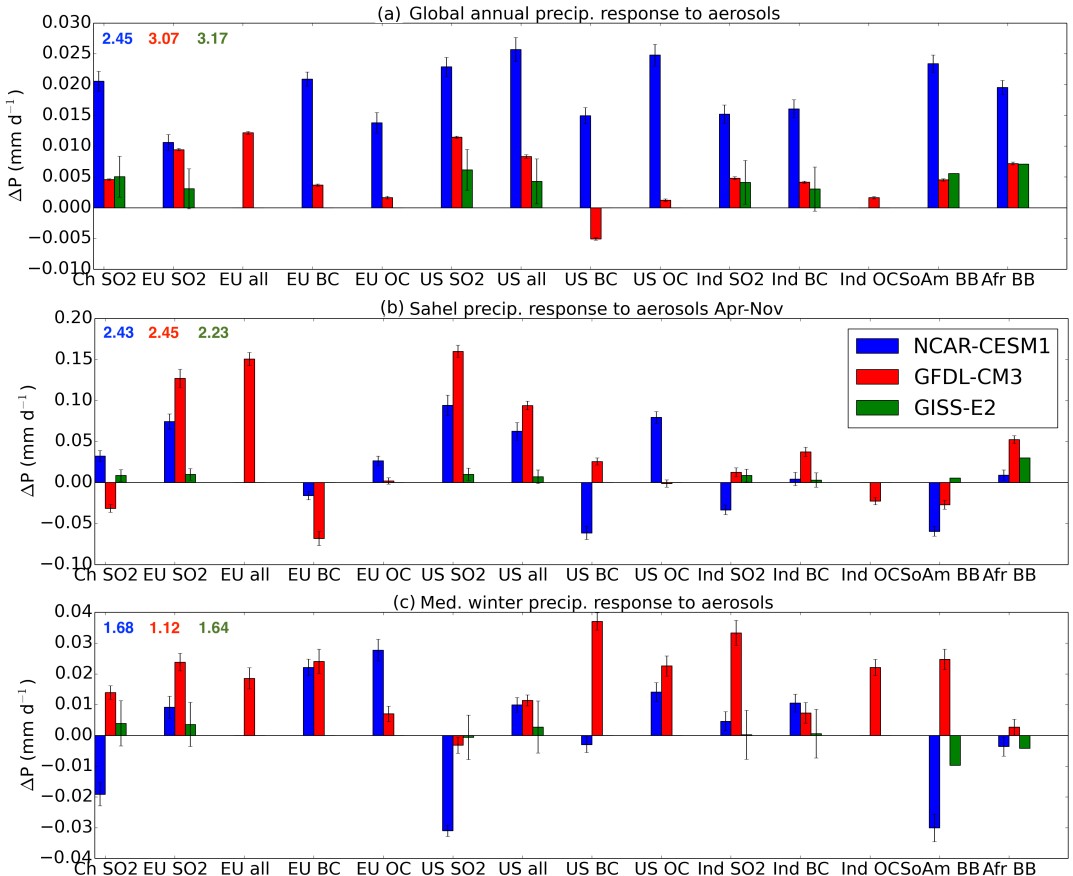

5  **Figure 2: Regional and global precipitation response to each individual aerosol emissions decrease (Table 1). (a) Global, annual (b) Sahel, Jun-Sep, (c) Mediterranean, Oct-Mar. Error bars represent ±1σ. Values in the upper left of each panel are control mean precipitation values for each region and time period for each model (green: GISS-E2, red: GFDL-CM3, blue: CESM1).**



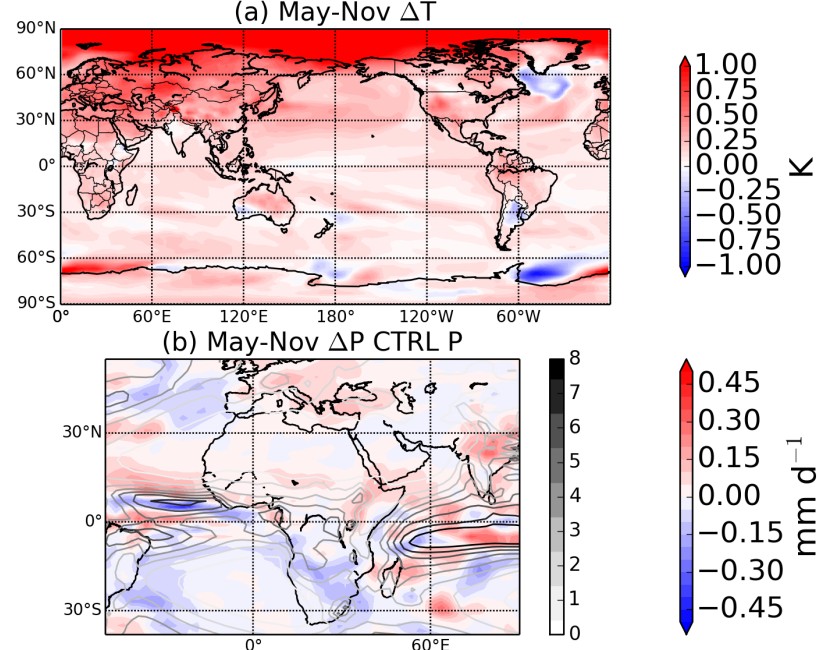

**Figure 3: Climate response in GFDL-CM3 to removal of European sulfur dioxide emissions. (a) Change in May-Nov mean surface temperature over a 200 year simulation (b) Change in mean May-Nov precipitation (colors). Control precipitation values shown in grey**





Figure 4: Scatterplot of Sahel precipitation change (June-Sep mean) due to aerosol regional emissions perturbations (symbols) and change in the interhemispheric temperature gradient in GFDL-CM3 (red), NCAR-CESM1 (blue), and GISS-E2 (green)



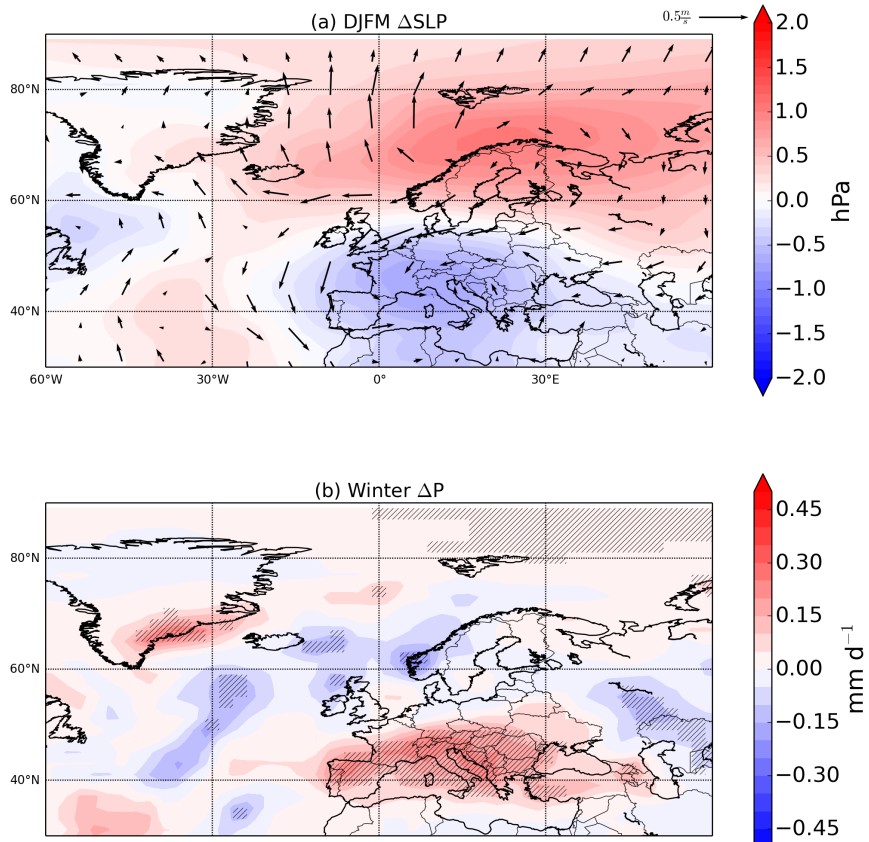

5 **Figure 5: Wintertime response in sea-level pressure and surface winds (a) and precipitation (b) to 80% reduction of European SO₂ emissions in GFDL-CM3. Hatching indicates statistical significance at the 95% confidence level.**



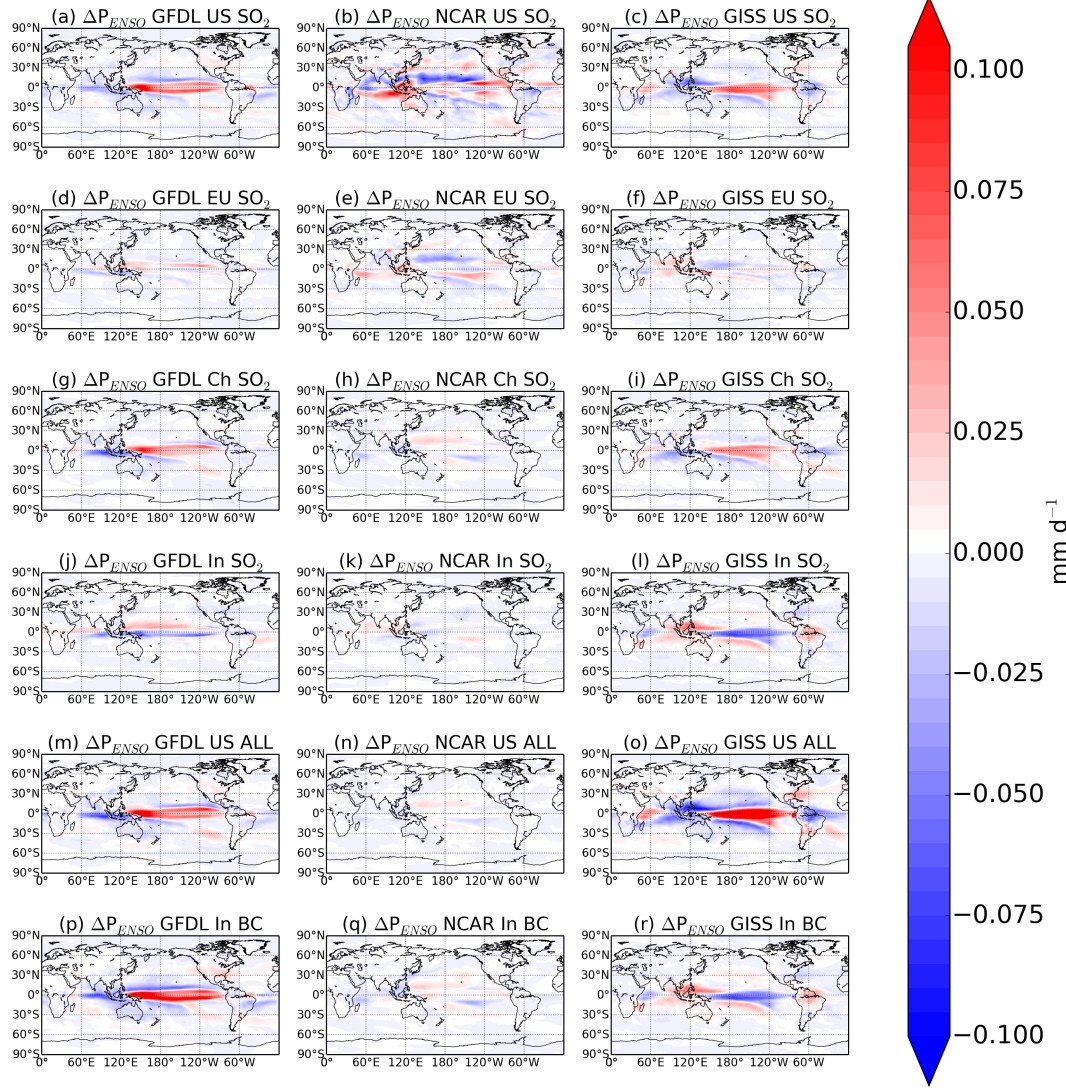

**Figure 6: 200-year annual mean ENSO component of the precipitation response to aerosol emissions decreases in each of the three models (GFDL-CM3, first column; NCAR-CESM1, second column; GISS-E2, third column) for several different regional emissions decreases (simulations indicated in figure titles; see Table 1). See text for methodology.**

