# Peer review of "Connecting regional aerosol emissions reductions to local and remote precipitation responses"

_Atmospheric Chemistry and Physics, 2018_

## Referee Comment (RC1) · Anonymous Referee #1 · 27 Jun 2018

Review of Westervelt et al. acp-2018-516

This is a high quality paper. The study is clean, thoroughly done, and I learned something. Bravo. I have a few minor comments.

Your final section is much heavier on the summary and pretty light on the conclusions. What else can you say about how this study fits more broadly into the scientific literature?

One important point about your analysis is that you did the step changes one at a time, which doesn't tell you about nonlinearity. I don't think you need to do any additional simulations, but it would be useful for you to comment (insofar as you're able to do so)

Printer-friendly version, Discussion paper buttons

about additivity of the perturbations, or lack thereof.

The fact that GISS-E2-R didn't include aerosol indirect effects is interesting in the light of Malavelle et al. (2018) [https://www.nature.com/articles/nature22974]. The conclusion from that paper is that the first aerosol indirect effect is far more important than the second one. This of course doesn't mean that GISS is "right", and the other two models are "wrong", but a comment may be in order.

The mechanism you invoke reminds me of a few papers: https://www.nature.com/articles/nclimate1857 https://agupubs.onlinelibrary.wiley.com/doi/full/10.1002/2014RG000449 https://agupubs.onlinelibrary.wiley.com/doi/full/10.1002/2015GL066903 All three of these support your mechanism (especially the first one) and might be useful to reference if appropriate.

One thing you don't mention about Sahelian rainfall is the character of the rain. Mean changes could indicate more extreme events. I don't know if this is relevant or if you can comment on it, given the scope of the study, but I thought I'd mention it.

Your discussion of Mediterranean precipitation changes is something of a counterpart to this paper: https://agupubs.onlinelibrary.wiley.com/doi/abs/10.1002/2017GL076669 I don't think you need to do anything to address this comment – just something interesting that occurred to me.

---

## Referee Comment (RC2) · Daniel M. Westervelt et al. · 9 Jul 2018

This study carries out extensive climate model simulations to understand the global and regional precipitation changes due to aerosol variations. Three models with different sophistications of aerosol effects are employed to provide an ensemble assessment. The model analysis is done in a quite comprehensive manner and the paper is well written overall. Therefore, I recommend accepting this manuscript by ACP after some necessary revisions as suggested below.

1) The limitations of current GCM in assessing aerosol effect on precipitation have to be clearly stated. For example, three GCM in this study do not account for the aerosol microphysical effects on convective clouds and precipitation which are still parameterized as the sub-grid scale processes (Wang et al., 2014, Fan et al., 2016).

2) Table 1, the sign of ERF from the removal of BC can be either positive or negative for different regions among three different models. Why is that? BC direct radiative forcing has been widely reported to be positive by previous modeling and observational studies (Ghan et al., 2012; Peng et al., 2016). Does your results imply the large spread of BC microphysical effects on cloud and precipitation among the models?

3) P3L18, are those model coupled with full chemistry? Like for CESM1, is the MOZART on?

4) P4L15-25 and Fig. S2, I'm not fully convinced that precipitation changes should be well correlated with ERF in physics. As you hinted in the paper, precipitation is related with atmospheric heating, while ERF is about the radiative flux variations at the top of atmosphere. The response of surface energy fluxes is an unknown factor. Moreover, I'm not sure if the global mean precipitation change is a good indicator here, as you have showed that the major spatial pattern of the simulated precipitation change is the "ENSO like" seesaw. The regional changes may be largely offset in the global mean.

5) P5L13-19, to better unravel the role of BC on convection, it would be useful to separately analyze the convective and stratiform precipitation in each model. I assume those two quantities are available for those models.

6) As the fully coupled models are used in this study, the simulated large-scale circulation changes should be closely linked with the polar climate change and the "Arctic amplification" is evident in Fig. 3. Therefore, the influence of emission changes on the Arctic sea ice and temperature should be relevant here, as discussed by Wang et al. (2018).

Suggested references:
- Wang, Y., M. H. Wang, R. Y. Zhang, S. J. Ghan, Y. Lin, J. X. Hu, B. W. Pan, M. Levy, J. H. Jiang, and M. J. Molina (2014), Assessing the effects of anthropogenic aerosols on Pacific storm track using a multiscale global climate model, P Natl Acad Sci USA, 111(19), 6894-6899.

- Fan, J. W., Y. Wang, D. Rosenfeld, and X. H. Liu (2016), Review of aerosol-cloud interactions: Mechanisms, significance, and challenges. J. Atmos. Sci., 73(11), 4221--4252.
- Ghan, S. J., X. Liu, R. C. Easter, R. Zaveri, P. J. Rasch, J. H. Yoon, and B. Eaton (2012), Toward a Minimal Representation of Aerosols in Climate Models: Comparative Decomposition of Aerosol Direct, Semidirect, and Indirect Radiative Forcing, Journal of Climate, 25(19), 6461-6476.
- Peng, J., et al. (2016), Markedly enhanced absorption and direct radiative forcing of black carbon under polluted urban environments, Proc Natl Acad Sci U S A, 113(16), 4266-4271.
- Wang, Y, J. Jiang J., H. Su, Y.-S. Choi, L. Huang, J. Guo, Y.-L. Yung (2018), Elucidating the role of anthropogenic aerosols in arctic sea ice variations. J. Clim., 31, 99–114.

---

## Author Comment (AC1) · 26 Jul 2018

**Responses to reviewer comments**

**Anonymous Referee #1**

This is a high quality paper. The study is clean, thoroughly done, and I learned something. Bravo. I have a few minor comments.

Your final section is much heavier on the summary and pretty light on the conclusions. What else can you say about how this study fits more broadly into the scientific literature?

We thank the reviewer for the kind words.

We have added some additional text placing our study into a broader context. For example, we mention a possible implication of our findings on aerosols and ENSO:

"More broadly, our findings suggest a possible anthropogenic influence on this mode of climate variability, which may complicate efforts to separate variability arising from naturally from those forced by anthropogenic drivers."

We also conclude the section by adding some additional text to the final paragraph:

"Aerosol-precipitation interactions remain one of the most uncertain aspects of future climate change, especially on the regional scale (Rosenfeld et al., 2008; Michibata et al., 2016). To reduce the uncertainty of how future regional aerosol decreases will impact regional precipitation, a thorough analysis with multiple models, including several regions and aerosol types, is needed. Our results show robust precipitation responses to regional aerosol emissions changes do occur, indicating promise for future work. One caveat of our study is that in each of the models, aerosols do not exert a microphysical effect on deep convective clouds; however they can alter precipitation associated with deep convection through the aerosol direct effect."

We have also edited text in other places throughout the conclusions section, with the aim to place our results in broader context. We refer the reviewer to the revised manuscript (with changes highlighted) for the full suite of revisions to this section.

One important point about your analysis is that you did the step changes one at a time, which doesn't tell you about nonlinearity. I don't think you need to do any additional simulations, but it would be useful for you to comment (insofar as you're able to do so) about additivity of the perturbations, or lack thereof.

A subset of our simulations could tell us something about additivity across different aerosol types within a given model. For example, the US_ALL simulation could be compared to a sum of US_SO2, US_BC, and US_OC. We have done that for US aerosols in GFDL-CM3 in Figure 1 below and find that summing of the individual forcers results in a much larger precipitation response than the combined perturbation simulation.

[Figure]

Figure 1: Sum of individual US aerosol forcing experiments compared to combined perturbation

In light of these results, we have added to following sentence to section 4.3 of the manuscript:

"In cases where regional aerosols were perturbed both individually and altogether (for example, US_ALL, US_SO2, US_OC, and US_BC), we find that the summation of the individual perturbations usually results in a larger precipitation response, both regionally and globally, compared to the combined perturbation (e.g. US_ALL), indicating nonlinearity among the individual responses (see Fig. 1 and Fig. S2)."

We have also added the following line of text to the conclusions:
"A possible avenue of further study may be combining different regions into a single perturbation simulation, resulting in a larger climate response and the ability to test for additivity or linearity among the simulations"

Our current suite of simulations does not provide information about additivity of the precipitation response across different regions. We leave this for future work.

The fact that GISS-E2-R didn't include aerosol indirect effects is interesting in the light of Malavelle et al. (2018) [https://www.nature.com/articles/nature22974]. The conclusion from that paper is that the first aerosol indirect effect is far more important than the second one. This of course doesn't mean that GISS is "right", and the other two models are "wrong", but a comment may be in order.

Correct, GISS-E2-R did not include the cloud lifetime effects or 'second' indirect effect. This is a good paper for us to cite. We have added the following sentence to the concluding discussion about the GISS results, which also partially address the reviewer's first comment about placing conclusions into a broader context:

"Using both climate models simulations and satellite observations of a major volcanic eruption, Malavelle et al. (2017) found that aerosol-induced changes in cloud liquid water path (the cloud lifetime effect) were undetectable, suggesting that the cloud lifetime effect may be less important than the cloud albedo effect for climate models."

The mechanism you invoke reminds me of a few papers:
https://www.nature.com/articles/nclimate1857
https://agupubs.onlinelibrary.wiley.com/doi/full/10.1002/2014RG000449
https://agupubs.onlinelibrary.wiley.com/doi/full/10.1002/2015GL066903 All three of
these support your mechanism (especially the first one) and might be useful to reference
if appropriate.

Each of these has been cited in the revised manuscript.

One thing you don't mention about Sahelian rainfall is the character of the rain. Mean
changes could indicate more extreme events. I don't know if this is relevant or if you can
comment on it, given the scope of the study, but I thought I'd mention it.

This is something we are currently working on for a future publication, so we prefer to
save comments about precipitation extremes for that manuscript.

Your discussion of Mediterranean precipitation changes is something of a counterpart to
this paper: https://agupubs.onlinelibrary.wiley.com/doi/abs/10.1002/2017GL076669 I
don't think you need to do anything to address this comment – just something interesting
that occurred to me.

We thank the reviewer for bringing this work to our attention.

---

## Author Comment (AC2) · 26 Jul 2018

**Responses to reviewer comments**

**Anonymous Referee #2**

This study carries out extensive climate model simulations to understand the global and regional precipitation changes due to aerosol variations. Three models with different sophistications of aerosol effects are employed to provide an ensemble assessment. The model analysis is done in a quite comprehensive manner and the paper is well written overall. Therefore, I recommend accepting this manuscript by ACP after some necessary revisions as suggested below.

We thank the reviewer for the comments.

1) The limitations of current GCM in assessing aerosol effect on precipitation have to be clearly stated. For example, three GCM in this study do not account for the aerosol microphysical effects on convective clouds and precipitation which are still parameterized as the sub-grid scale processes (Wang et al., 2014, Fan et al., 2016).

We have added the following sentence to the concluding paragraph of the manuscript:

"One caveat of our study is that in each of the models, aerosols do not exert a microphysical effect on deep convective clouds; however they can alter precipitation associated with deep convection through the aerosol direct effect."

Further model description relevant to precipitation and clouds is referenced in Westervelt et al. (2017).

Westervelt, D.M., A.J. Conley, A.M. Fiore, J.-F. Lamarque, D. Shindell, M. Previdi, G. Faluvegi, G. Correa, and L.W. Horowitz, 2017: Multimodel precipitation responses to removal of U.S. sulfur dioxide emissions. *J. Geophys. Res. Atmos.*, **122**, no. 9, 5024-5038, doi:10.1002/2017JD026756.

2) Table 1, the sign of ERF from the removal of BC can be either positive or negative for different regions among three different models. Why is that? BC direct radiative forcing has been widely reported to be positive by previous modeling and observational studies (Ghan et al., 2012; Peng et al., 2016). Does your results imply the large spread of BC microphysical effects on cloud and precipitation among the models?

GFDL-CM3 and GISS-E2 only includes direct effects for BC, thus removing BC in results in small negative ERF values in these cases. In the case of positive numbers for NCAR-CESM1, this could be caused by the differences in aerosol treatment between the models. For instance, CESM1 uses an internal mixing approach with modal aerosol microphysics. Internally mixed BC-sulfate particles can activate clouds in this model setup, which could lead to a slight positive radiative forcing when BC is removed. Another possible explanation is that since these regional BC perturbations can be quite

small in magnitude, the role of internal climate variability may be outweighing the BC forcing, especially for a global mean ERF value.

Finally, many of the ERFs reported in Table 1 are close to zero and are not statistically significantly different from zero, so the signs of these small numbers should not be overanalyzed. For example, the standard error for the IN_BC in GISS-E2 simulation is 0.028 W m$^{-2}$, so the ERF mean of 0.011 W m$^{-2}$ is not even significant at the 1-sigma level. Similarly, none of the BC ERF values in GFDL-CM3 are significant at the 2-sigma level. We have edited Table 1 in the revised manuscript, putting the ERFs that are significant at the 2-sigma level in **boldface** type.

We have also added the following sentence to the manuscript in light of the reviewer comments:
"The black carbon aerosol global mean ERF (Table 1) varies in sign and magnitude, indicating a strong sensitivity to different model configurations for black carbon and, perhaps, a role for internal climate variability. In many of the black carbon simulations, the global mean aerosol ERF values reported are not statistically significant."

3) P3L18, are those model coupled with full chemistry? Like for CESM1, is the MOZART on?

Yes, all models include full chemistry, as stated in the manuscript on Page 3, Line 17-20.

4) P4L15-25 and Fig. S2, I'm not fully convinced that precipitation changes should be well correlated with ERF in physics. As you hinted in the paper, precipitation is related with atmospheric heating, while ERF is about the radiative flux variations at the top of atmosphere. The response of surface energy fluxes is an unknown factor. Moreover, I'm not sure if the global mean precipitation change is a good indicator here, as you have showed that the major spatial pattern of the simulated precipitation change is the "ENSO like" seesaw. The regional changes may be largely offset in the global mean.

We agree with the reviewer that the correlation between ERF and global precipitation is imperfect, and do not intend to imply a strong causal relationship. Therefore, based on the reviewer comment we have removed the following sentence from the manuscript (and a similar one in the conclusions section) in order to not overemphasize a causal relationship between ERF and global precipitation:

 "This suggests that TOA aerosol ERF may explain some of the variation in global precipitation response, but not all of it."

We have also added the following sentence regarding a caveat to using global precipitation:

"Global precipitation may also be an imperfect metric for correlation, if opposite-signed regional changes are largely offset in the global mean."

We prefer to keep Fig. S2 in the supplemental section, however, as this figure allows for comparison between similar studies, such as the work from PDRMIP, which is cited in our manuscript.

5) P5L13-19, to better unravel the role of BC on convection, it would be useful to separately analyze the convective and stratiform precipitation in each model. I assume those two quantities are available for those models.

We have looked at convective and large-scale precipitation responses to black carbon in the models. The figures below shows the total precipitation response, the large-scale response, the convective response, and the shallow convective response to zero-out India BC emissions in GFDL-CM3 and NCAR-CESM1. As can be seen in the figures, both the large-scale and convective responses exhibit large amounts of noise and variability. Convective precipitation seems to dominate the total response, especially in convective regions such as the tropics. Large-scale precipitation responds more strongly in the mid-latitudes.

[Figure]

Figure 2: Total, large-scale, convective, and shallow convective precipitation response to zero India BC emissions in GFDL-CM3

[Figure]

Figure 3: Large-scale and convective precipitation response to zero India BC emissions in NCAR-CESM1

Because of large amounts of variability and lack of statistical significance, it is difficult to discern anything further from the breakdown of precipitation types that cannot already

be discerned from the total precipitation. Thus, we elect to keep the discussion in the paper as is and leave these figures in the response to reviews document.

6) As the fully coupled models are used in this study, the simulated large-scale circulation changes should be closely linked with the polar climate change and the "Arctic amplification" is evident in Fig. 3. Therefore, the influence of emission changes on the Arctic sea ice and temperature should be relevant here, as discussed by Wang et al. (2018).

In the revised manuscript, we have cited the Wang et al. (2018) paper in our discussion of Figure 3.